# Cutting-Force Modeling Study on Vibration-Assisted Micro-Milling of Bone Materials

**DOI:** 10.3390/mi14071422

**Published:** 2023-07-14

**Authors:** Peng Shang, Huaiqing Zhang, Xiaopeng Liu, Zhuang Yang, Bingfeng Liu, Teng Liu

**Affiliations:** School of Mechanical Engineering, Hebei University of Technology, Tianjin 300400, China; 202121202066@stu.hebut.edu.cn (H.Z.); 18735435669@163.com (X.L.); 202021202056@stu.hebut.edu.cn (Z.Y.); 202231205119@stu.hebut.edu.cn (B.L.)

**Keywords:** micro-milling, bone materials, vibration-assisted, cutting force, anisotropy

## Abstract

This study aims to enhance surgical safety and facilitate patient recovery through the investigation of vibration-assisted micro-milling technology for bone-material removal. The primary objective is to reduce cutting force and improve surface quality. Initially, a predictive model is developed to estimate the cutting force during two-dimensional (2D) vibration-assisted micro-milling of bone material. This model takes into account the anisotropic structural characteristics of bone material and the kinematics of the milling tool. Subsequently, an experimental platform is established to validate the accuracy of the cutting-force model for bone material. Micro-milling experiments are conducted on bone materials, with variations in cutting direction, amplitude, and frequency, to assess their impact on cutting force. The experimental results demonstrate that selecting appropriate machining parameters can effectively minimize cutting force in 2D vibration-assisted micro-milling of bone materials. The insights gained from this study provide valuable guidance for determining cutting parameters in vibration-assisted micro-milling of bone materials.

## 1. Introduction

Bone-cutting surgery is an important surgical method for bone-material removal in clinical practice [1]. As a natural composite material, bone material has complex structural characteristics, such as heterogeneity and hard brittleness [2]. Therefore, bone-cutting surgery can be regarded as a precision machining process. As an essential processing method, micro-milling has the advantages of high material-removal rate [3] and good surface quality so that it can be used in bone-material removal surgery. However, the mechanical load and temperature in the micro-milling process have a great influence on the milling of bone materials [4]. Excessive cutting force will cause serious damage to bone tissue, and it will also produce large cutting heat to increase the temperature and, thus, affect the biological activity of bone tissue. Therefore, in this study, 2D vibration-assisted machining technology is applied to the micro-milling process of bone materials. The purpose is to reduce the cutting force of bone materials during surgery and obtain better surface quality. Therefore, the accurate modeling of the cutting force of 2D vibration-assisted micro-milling of bone materials can help doctors select appropriate processing parameters and surgical tools, and achieve precise control of the cutting process with the purpose of reducing bone-tissue necrosis and reducing postoperative recovery time.

Currently, most of the research on the cutting-force model of bone materials focuses on the homogeneous model and the bone-drilling force model. TSAI et al. [5] calculated the drilling force and torque in the drilling process according to metal-cutting theory so as to provide accurate prediction values for surgery and training. Lee [6] et al. proposed a model that can predict the drilling force and torque during bone drilling. The model comprehensively considers factors such as the geometry of the drill bit, cutting conditions, and friction characteristics. The accuracy of the model is well matched with the experiment through the drilling experiment of bovine tibial cortical bone. Mitsuishi et al. [7] evaluated the relationship between different cutting conditions and the cutting characteristics of cortical bone and cancellous bone by establishing the relationship model between cutting force and different undeformed chip thickness, cutting speed, and other factors when milling cortical bone and cancellous bone. Liao et al. [8] proposed a new milling-force model based on the special structure and characteristics of bone tissue. The model considers the direction of the bone unit, the geometry of the tool, the ploughing effect, and the radial runout, and predicts the cutting force coefficient. It can be better used to assist robotic surgery, optimize cutting parameters, and guide the design of orthopedic surgical tools. Liu [9] obtained the influence of spindle speed, feed rate, and cutting depth on cutting force through cortical bone milling experiments. It was found that as the spindle speed increased, the cutting force decreased, while the feed rate and cutting depth showed a positive correlation. Liu obtained the cutting-force prediction model through the measured cutting-force data and the least-squares regression method, and verified the validity of the model through experimental design.

Vibration-assisted machining technology is commonly employed for precision materials that are challenging to machine, such as titanium alloys [10] and high-temperature alloys, owing to its unique material-removal mechanism. Currently, the investigation of cutting forces in vibration-assisted machining of bone materials primarily revolves around vibration-assisted turning and vibration-assisted drilling. However, there is limited research on the cutting force involved in 2D vibration-assisted micro-milling. Alam et al. [11] applied ultrasonic vibration technology to the drilling of bone materials. By building an ultrasonic vibration-drilling platform for experiments, it was found that, compared with conventional drilling methods, ultrasonic vibration-drilling of bone materials can significantly reduce the cutting force and torque, make the chips easier to discharge, and reduce the surface damage of bone materials. Sugita et al. [12,13,14] of Tokyo University in Japan, applied vibration to the tool on the basis of the original bone-cutting technology. By observing and analyzing the cracks generated by the bone material during the cutting process, they found that applying vibration can change the crack propagation mode so that the crack can penetrate the bone unit, prevent the large-scale fracture of the bone material, and form a regular stable chip, which leads to improved removal efficiency of the bone material, and the vibration-cutting technology can effectively reduce the cutting force so that the bone material can be efficiently removed under low load. Wei Bai [15,16] proposed the cutting-force model of 2D elliptical vibration-cutting process, and analyzed the shear angle of different friction regions by using the principle of minimum energy. Based on the change of transient cutting thickness and transient shear angle during the cutting process, the cutting force of elliptical vibration cutting in a single cycle was obtained. He also analyzed the cutting performance and cutting mechanism of bone material in vibration cutting, and compared the cutting forces under the two cutting methods. The results show that the strain rate of bone material increases during vibration-assisted cutting, and the crack is easy to propagate through the bone unit. Finally, the chip shape generated during vibration-assisted cutting of bone material is stable, the cutting force is reduced, and the cracks and defects of the machined surface are generated.

In summary, the current vibration-assisted cutting methods for bone materials are also focused on vibration-assisted turning and vibration-assisted drilling. There are few studies on 2D vibration-assisted milling of bone materials, and how the vibration parameters affect the cutting force of bone materials in the milling process still needs to be comprehensive. Therefore, it is of great significance to study the application of vibration-assisted technology in micro-milling of bone materials to optimize the processing effect, reduce the trauma of bone materials, and improve the recovery speed.

## 2. Mechanical Properties of Bone Materials

Bone material can be seen as a natural fiber-reinforced composite material, and its microstructure is shown in Figure 1a. Cortical bone is composed of interstitial bone, osteon, bone cement line, and haversian canal, and its main component is composed of osteon and interstitial bone. Bone materials can be regarded as orthotropic materials. The mechanical properties of bone material have been studied by some scholars under tensile and compressive loads. However, little research has been conducted on its shear properties. In addition, the nature of the cutting process is the deformation of the cutting layer material under the shearing action of the tool. Therefore, studying the shear properties of bone materials is essential for understanding the process of vibration-assisted milling of bone materials.

Due to the difficulty of using human bone material samples and the fact that researchers have compared the properties of human and animal bone materials through many tests, it was found that the mechanical properties of bovine bone are similar to those of humans [17,18,19,20]. Therefore, this study selected bovine bone, which has properties close to human bone, for shear tests. Fresh bovine femur bones were selected to remove the soft tissues, such as muscles. Due to the inhomogeneous composition and microstructure of the bones, which led to considerable differences between the mechanical properties of different locations in the axial direction and different regions in the circumferential direction, the samples were obtained from the regions with uniform wall thickness on the bones as much as possible. According to the structural properties of the bone material, the sample preparation of the material was divided into two types: for the parallel sample preparation, the geometry of each sample was 32 mm × 5 mm × 3 mm. For the cross and vertical sample preparation, each sample’s geometry was 38 mm × 20 mm × 3 mm. The three samples with the loading direction during shearing are shown in Figure 1. The samples were prepared on a VMC850 machining center under water-cooled conditions, stored in saline, and frozen at −15 °C to keep the mechanical properties of the bone material as constant as possible until thawing before testing.

The bone material samples were placed in the shear fixture, and the fixture was placed on the working table of the testing machine. The samples were sheared along different directions of the bone material at a loading speed of 0.5 mm/min. After the test, the test results were output by the computer. The shear strength $\tau_{s}$ of bone material can be calculated by Equation (1).
(1)$$\tau_{s}=\frac{F_{max}}{2S}$$
where ${\;F}_{max}$ is the maximum shear load and S is the cross-sectional area of the original shear surface of the sample.

The stress–strain curve obtained based on the shear test of bone material is shown in Figure 2. When shearing along the parallel direction and cross direction of the bone material sample, the stress–strain curve increases linearly in the early stage, but, in the later stage, the growth rate of stress with strain gradually decreases. However, when shearing along the bone material specimen’s vertical direction, the initial stage’s stress growth rate is slow, and it only maintains a linear growth state in the middle stage. It shows linear elasticity and elastoplasticity, indicating that the bone material has apparent anisotropy. The shear strength of the bone material in three directions is shown in Table 1.

## 3. Vibration-Assisted Micro-Milling Force Modeling

### 3.1. Anisotropic Cutting-Force Modeling of Bone Materials

This study studied the vibration milling process of bone materials using a two-blade flat-end milling cutter. The rectangular co-ordinate system shown in Figure 3 was established. X is the feed direction of the workpiece, Y is the groove width direction, and Z is the groove depth direction. The vibration platform drives the bone material workpiece to vibrate in X and Y directions. In this study, the vibration platform can apply a continuously adjusted vibration frequency of 0~4000 Hz and a continuously adjusted amplitude of 0~3 μm in the X and Y directions, respectively.

Since the axial force in the milling process is much smaller than the cutting force in the X and Y directions, this study mainly analyzes the cutting force of the bone material in the XOY plane. As shown in Figure 4a, in the milling process, the cutting edge of the milling cutter can be regarded as a plurality of discrete oblique cutting micro-element layers in the axial direction. The cutting force applied to the bone material workpiece by each cutting edge micro-element can be decomposed into the radial force $dF_{r}\;$along the radial direction of the tool, the tangential force$\;dF_{t\;}$perpendicular to the radial direction, and the axial force $dF_{a\;}$ along the axial direction of the tool. Each cutting component can be decomposed into two parts: the shear force generated by the shear effect and the ploughing force caused by the friction effect [21]. Therefore, in the XOY plane, the radial force$\;dF_{r\;}$can be decomposed into radial shear force$\;dF_{rc}\;$and radial ploughing force $dF_{rp}$, and the tangential force $dF_{t}\;$can be decomposed into tangential shear force $dF_{tc\;}$ and tangential ploughing force $dF_{tp}$, as shown in Figure 4b. Here, *H* represents the axial cutting depth, $d_{Z}\;$represents the axial height of the milling cutter infinitesimal, and θ represents the instantaneous tooth angle of the cutting edge.

Therefore, according to the principle of oblique cutting, the relationship between tangential force $dF_{j,t}^{i}$, radial force $dF_{j,r}^{i}$, axial force $dF_{j,a}^{i}$, and instantaneous cutting thickness $h_{i}(\theta ,j),\;$and milling depth dz is shown in Equations (2) and (3).
(2)$$\left(\begin{array}{c} dF_{j,t}^{i}\\ dF_{j,r}^{i}\\ dF_{j,a}^{i} \end{array}\right)=\left(\begin{array}{c} dF_{j,tc}^{i}+dF_{j,tp}^{i}\\ dF_{j,rc}^{i}+dF_{j,rp}^{i}\\ dF_{j,ac}^{i}+dF_{j,ap}^{i} \end{array}\right)=\left(\begin{array}{c} K_{tc}^{i}\\ K_{rc}^{i}\\ K_{ac}^{i} \end{array}\right)h_{i}(\theta ,j)dz+\left(\begin{array}{c} \begin{array}{c} K_{tp}^{i}\\ K_{rp}^{i} \end{array}\\ K_{ap}^{i} \end{array}\right)dz$$
(3)$$dz=\frac{r\cdot d\theta }{\tan\lambda }$$
where *i* is the cutting direction, *i* = 1, 2, 3 means parallel, cross, and vertical cutting directions, respectively; *j* is the tooth number, *j* = 0, 1; $dF_{j,t}^{i}$, $dF_{j,r}^{i}$, $dF_{j,a}^{i}$ are the tangential, radial, and axial cutting forces of the *j*th tooth-cutting micro-element in the corresponding cutting direction; $dF_{j,tc}^{i}$, $dF_{j,rc}^{i}$, $and\;dF_{j,ac}^{i}$ are the tangential, radial, and axial shear forces of the *j*th tooth-cutting micro-element in the corresponding cutting direction; $dF_{j,tp}^{i}$, $dF_{j,rp}^{i}$, and $dF_{j,ap}^{i}$ are the tangential, radial, and axial ploughing forces of the *j*th tooth micro-element in the corresponding cutting direction; $K_{tc}^{i}$, $K_{rc}^{i}$, and $K_{ac}^{i}$ are the tangential, radial, and axial shear force coefficients in the corresponding cutting direction; $K_{tp}^{i}$, $K_{rp}^{i}$, and $K_{ap}^{i}$ are the tangential, radial, and axial ploughing force coefficients in the corresponding cutting direction; *θ* is the instantaneous tooth position angle; *h_i_*(*θ*, *j*) is the cutting thickness of the *j*th tooth in the corresponding cutting direction when the tooth position angle is *θ*; d*z* is the axial height of the milling micro-element; *r* is the radius of the milling cutter; and *λ* is the edge inclination angle of the milling cutter; and it is generally considered that the edge inclination angle of the end mill is equal to the helix angle.

Therefore, the instantaneous cutting force on the bone material workpiece in the feed direction (X-direction), normal direction (Y-direction), and axial direction (Z-direction) can be expressed by Equation (4).
(4)$$\left\{\begin{array}{l} F_{x}(\theta )=\frac{r}{\tan\lambda }\sum_{1}^{N}\int {}_{\theta_{\mathrm{st}}}^{\theta_{\mathrm{ex}}}\left(dF_{j,t}^{i}(\theta )\cos\theta +dF_{j,r}^{i}(\theta )\sin\theta \right)d\theta \\ F_{y}(\theta )=\frac{r}{\tan\lambda }\sum_{1}^{N}\int {}_{\theta_{\mathrm{st}}}^{\theta_{\mathrm{ex}}}\left(-dF_{j,t}^{i}(\theta )\sin\theta +dF_{j,r}^{i}(\theta )\cos\theta \right)d\theta \\ F_{z}(\theta )=\frac{r}{\tan\lambda }\sum_{1}^{N}\int {}_{\theta_{\mathrm{st}}}^{\theta_{\mathrm{ex}}}\left(-dF_{j,a}^{i}(\theta )\right)d\theta \end{array}\right.$$
where$\;F_{x}(\theta )$, $F_{y}(\theta )$, and $F_{z}(\theta )$ are the cutting forces of the bone material workpiece in X, Y, and Z directions; $\theta_{st}$, $\theta_{ex}$ is the angle of the milling integration interval; and *N* is the number of cutting edges of the milling cutter, and, for two-edge milling cutter, *N* = 2.

Due to the helix angle of the flush-end mill selected for this project, the cutting-edge micro-element at the axial depth of cut *z* lags behind the cutting-edge micro-element at the bottom of the milling cutter during the milling process, and the lag angle ${\;\theta }_{z\;}$ at the axial depth of cut *z* can be expressed by Equation (5).
(5)$$\theta_{z}=\frac{\tan\lambda \cdot z}{r}$$

Therefore, the upper- and lower-limit angles of the spiral groove of the milling cutter in the axial integration interval and the range of values are shown in Table 2.

### 3.2. Shear Force Coefficient Model

The shear force coefficient reflects the shearing action of the tool on the workpiece, and its numerical magnitude depends mainly on the material’s physical properties and the tool’s geometric parameters. Since the milling process of a helical milling cutter can be considered a bevel-cutting process [22,23], according to the shear theory proposed by Armarego [24], the shear force coefficient can be expressed by Equation (6).
(6)$$\left\{\begin{array}{c} K_{tc}^{i}=\frac{\tau_{i}}{\sin\varphi_{i}}\frac{\cos(\beta_{n}-\gamma_{n})+\tan\eta_{c}\sin\beta_{n}\tan\lambda }{\sqrt{\cos^{2}(\varphi_{i}+\beta_{n}-\gamma_{n})+\tan^{2}\eta_{c}\sin^{2}\beta_{n}}}\\ K_{rc}^{i}=\frac{\tau_{i}}{\sin\varphi_{i}\cos\lambda }\frac{\sin(\beta_{n}-\gamma_{n})}{\sqrt{\cos^{2}(\varphi_{i}+\beta_{n}-\gamma_{n})+\tan^{2}\eta_{c}\sin^{2}\beta_{n}}}\\ K_{ac}^{i}=\frac{\tau_{i}}{\sin\varphi_{i}}\frac{\cos(\beta_{n}-\gamma_{n})\tan\lambda -\tan\eta_{c}\sin\beta_{n}}{\sqrt{\cos^{2}(\varphi_{i}+\beta_{n}-\gamma_{n})+\tan^{2}\eta_{c}\sin^{2}\beta_{n}}} \end{array}\right.$$
where *τ_i_* is the shear strength of the bone material in the corresponding cutting direction, and its value is shown in Table 1; $\beta_{i}$ is the friction angle in the corresponding cutting direction, which can be obtained from $\tan\beta_{i}=\mu_{i}$; $\mu_{i}$ is the friction coefficient between the front surface of the tool and the bone material in the corresponding cutting direction; $\gamma_{n}\;$is the nominal rake angle of the tool; $\eta_{c}\;$is the chip flow angle; and *φ_i_* is the normal shear angle of the bone material in the corresponding cutting direction.

According to the chip flow principle, the chip flow angle equals the edge inclination angle of bevel cutting, which can be approximated as *η_c_* = *λ*. The shear angle can be calculated by Equation (7).
(7)$$\varphi_{i}=\frac{\pi }{4}-\frac{\beta_{i}-\gamma_{n}}{2}$$

The friction coefficients at various cutting directions can be obtained from Table 3.

### 3.3. Ploughing Force Coefficient Model

In the conventional milling method, the ploughing force generated by the ploughing phenomenon is much smaller than the shear force generated by the shear action because the feed per tooth is taken as a large value compared to the radius of the tool tip arc, so only the shear action is often considered in micro-milling machining. However, in the micro-milling process, the feed per tooth and the blunt radius of the tool are of the same micron scale, and the ploughing force is a larger proportion of the cutting force due to the scale effect, which cannot be neglected [26]. Additionally, the smaller the cutting thickness, the larger the proportion of ploughing force in the cutting force, and its magnitude can be expressed by Equation (8).
(8)$$F_{p}^{i}=b\tau_{\mathrm{MN}}^{i}l_{\mathrm{MN}}=b\tau_{i}\cos(2\eta_{i})l_{\mathrm{MN}}$$
where ${\;F}_{p}^{i}\;$is the ploughing force of the bone material in the corresponding cutting direction; $b_{i}\;$is the cutting width in the corresponding cutting mode; $\tau_{\mathrm{MN}}^{i}\;$is the frictional shear stress of the bone material along the cutting edge in the corresponding cutting mode; $l_{\mathrm{MN}}^{i}\;$is the length of the ploughing area in the corresponding cutting mode; $\eta_{i}\;$is the angle between the cutting slip line and the bottom cutting surface in the corresponding cutting mode; and $l_{\mathrm{MN}}^{i}\;$and ${\;\eta }_{i\;}$ can be obtained from Equations (9) and (10), respectively, according to the slip theory proposed by Waldorf.
(9)$$l_{\mathrm{MN}}^{i}=\frac{r_{e}}{\sin\eta_{i}}$$
(10)$$\eta_{i}=0.5\cos^{-1}(\mu_{i})$$
where $r_{e}$ is the radius of the blunt circle of the tool.

Therefore, the plow force coefficient can be expressed by Equation (11).
(11)$$\left\{\begin{array}{c} K_{tp}^{i}=F_{p}^{i}\frac{\sin\gamma_{e}}{b_{i}}=\tau_{i}cos(2\eta_{i})l_{\mathrm{MN}}^{i}sin\gamma_{e}\\ K_{rp}^{i}=F_{p}^{i}\frac{\cos\gamma_{e}}{b_{i}}=\tau_{i}cos(2\eta_{i})l_{\mathrm{MN}}^{i}cos\gamma_{e} \end{array}\right.$$
where $\gamma_{e}$ is the equivalent rake angle.

During micro-milling, the geometrical structure parameters of the tool are changed due to the presence of the blunt circle radius of the tool, mainly in the form of a change in the value of the rake angle of the tool, especially when the cutting thickness is close to or smaller than the blunt circle radius of the tool, and the tool edge shape can be considered as a tool with a large negative rake angle, whose equivalent rake angle can be obtained by the calculation of Equation (12) [27].
(12)$$\gamma_{e}=\left\{\begin{array}{c} arccos\left(\frac{r_{e}-h_{i}(\theta ,j)}{r_{e}}\right)-\frac{\pi }{2}\;\;h_{i}(\theta ,j)\leq r_{e}(1+sin\gamma_{n})\\ \gamma_{n}\;\;\;\;\;\;\;\;{\;\;h}_{i}(\theta ,j)>r_{e}(1+sin\gamma_{n}) \end{array}\right.$$

This study used SGOS550 superfine tungsten steel-coated two-edge milling cutter from Taiwan, and the milling cutter parameters are shown in Table 4.

### 3.4. Instantaneous Cutting Thickness Model

From the above analysis, it is clear that the results of the calculation of the instantaneous cutting thickness are related to the accuracy of the measure of the cutting force of the bone material [28]. In contrast, during vibration-assisted milling, there is intermittent cutting processing between the milling tool and the workpiece, which significantly changes the processing method. In this section, the vibration parameters and the eccentricity effect of the tool during vibration-assisted micro-milling of bone material are considered comprehensively to analyze and calculate the instantaneous cutting thickness during cutting.

In this study, a two-edged flush-end mill was used to mill bone material, and, due to errors in the manufacturing or installation of the milling tool, the tool eccentricity caused runout of the milling tool in the radial direction, which affected the calculation of the milling tool trajectory, cutting thickness, and cutting force [29,30]. Therefore, the locus of the tool tip relative to the workpiece during vibration-assisted bone material milling consists of four superimposed parts in the X direction, which are the feed motion of the workpiece, the rotational motion of the tool, the vibration of the workpiece, and the radial runout of the tool, while in the Y direction, it consists of the rotational motion of the tool, the vibration of the workpiece, and the radial runout of the tool. The motion trajectory of the tool tip on the workpiece can be expressed by Equation (13).
(13)$$\left\{\begin{array}{c} x=vt+r\;sin\left(\omega t-\frac{2\pi j}{N}\right)+r_{0}\;sin\left(\omega t+\psi \right)+A_{x}\;sin\left(2\pi f_{x}t+\varphi_{x}\right)\\ y=rcos\left(\omega t-\frac{2\pi j}{N}\right)+r_{0}\;cos\left(\omega t+\psi \right)+A_{y}\;sin\left(2\pi f_{y}t+\varphi_{y}\right) \end{array}\right.$$
where *x* and *y* are the co-ordinates of the tool tip relative to the workpiece in the X and Y directions, respectively; *ω* is the angular velocity; *t* is the machining time; *v* is the milling tool feed speed; *A_x_* and *A_y_* are the amplitudes of the vibration platform in the X and Y directions; *f_x_* and *f_y_* are the frequencies of the vibration platform in the X and Y directions, respectively; *φ_x_* and *φ_y_* are the initial phase angles of the vibration signal source in the X and Y directions; $\gamma_{0}$ is the milling tool eccentricity; and *ψ* is the milling tool eccentricity angle.

In the conventional milling process, the trajectory of the tool tip relative to the workpiece is a smooth sub-pendulum line synthesized by the rotational motion of the tool and the feed motion of the workpiece, as can be seen from Figure 5. Since this study uses a 2D vibration platform to apply vibration effects on the bone material workpiece after the 2D vibration is applied, the relative motion trajectory of the tool tip on the workpiece is also affected by the phase difference, and frequency and amplitude of the 2D vibration signal source, etc. The relative motion trajectory of the tool tip on the workpiece is a continuously oscillating curve.

The accuracy of the cutting thickness calculation results is crucial to the modeling of the milling forces of bone materials. In 2D vibration-assisted micro-milling, the instantaneous cutting thickness generated by the cutting edge is, thus, complicated by the constant oscillation of the relative motion trajectories of the tool teeth on the workpiece. The instantaneous cutting thickness is no longer composed of the motion trajectories of two adjacent tool tips, but is formed by the interlocking cutting trajectories of the interrelated tool tips, and the vibration effect causes the milling process to become an intermittent cutting process in which the vibration effect leading to periodic separation of the tool from the workpiece and the generation of empty cutting in the radial range of the cutting edge.

Due to the effect of 2D vibrations, four cases of cutting thickness of bone material exist depending on whether the current tip of the cutter lies outside the maximum contour line already formed by the teeth:(1)Different cutting edges lead to empty cutting: when the tool center is located in O_1_ position, the current tool tip position B_1_ falls within the cutting trajectory already formed by the previous tool tip. Currently, the cutting edge is not involved in cutting and the cutting thickness is zero;(2)Chip formed by different cutting edges: when the tool center is located in O_2_ position, the current tool tip position B_2_ falls outside the cutting trajectory already formed by the previous tool tip and the instantaneous cutting thickness is A_2_B_2_;(3)The same cutting edge leads to empty cutting: when the tool center is located in the O_3_ position, the current tip position B_4_ falls within the cutting trajectory already formed by the tip. Currently, the cutting edge does not participate in cutting and the cutting thickness is zero;(4)Chip formed by the same cutting edge: when the tool center is located in O_4_ position, the current tool tip position B_6_ falls outside the cutting trajectory already formed by the tool tip and the instantaneous cutting thickness is B_5_B_6_.

If the current tool tip is located outside the maximum contour line already formed by the tool tooth, the instantaneous cutting thickness is calculated for the above two cases (2) and (4). Since the current tool center O trajectory, the point P_s_(*x*_s_, *y*_s_) on the previous cutting edge trajectory and the point P(*x*, *y*) on the current cutting edge trajectory is on the same line and the following linear equation is available.
(14)$$\frac{x_{o}-x}{y_{o}-y}=\frac{x-x_{s}}{y-y_{s}}$$

The tool center trajectory Equation is:(15)$$\left\{\begin{array}{c} x_{o}=vt+r_{0}\sin\left(\omega t+\psi \right)+A_{x}\sin(2\pi f_{x}t+\varphi_{x})\\ y_{o}=r_{0}\cos\left(\omega t+\psi \right)+A_{y}\sin(2\pi f_{y}t+\varphi_{y}) \end{array}\right.$$

From the above analysis, it can be seen that there are two cases in the calculation process of cutting thickness, as shown in Figure 6a and 6b, respectively. Therefore, for the calculation of cutting thickness should first determine whether the previous tool tip trajectory P_s_(*x*_s_, *y*_s_) falls on the previous tool tip motion trajectory or the current tool tip motion trajectory.

When P_s_ falls on the previous tool tip trajectory, the instantaneous cutting thickness is formed by the trajectory of different tool teeth. Its instantaneous cutting thickness model is shown in Figure 6a and the relationship between the trajectory of P_s_ point and its corresponding cutting time *t*_s_ can be expressed by Equation (16).
(16)$$\left\{\begin{array}{l} x_{s}=vt_{s}+t\sin\left(\omega t_{s}-\frac{2\pi \left(j-1\right)}{N}\right)+r_{0}\sin\left(\omega t_{s}+\psi \right)+A_{x}\sin\left(2\pi f_{x}t_{s}+\varphi_{x}\right)\\ y_{s}=r\cos\left(\omega t_{s}-\frac{2\pi \left(j-1\right)}{N}\right)+r_{0}\cos\left(\omega t_{s}+\psi \right)+A_{y}\sin\left(2\pi f_{y}t_{s}+\varphi_{y}\right) \end{array}\right.$$

When P_s_ falls on the current tool tip trajectory, the instantaneous cutting thickness is formed by the motion trajectory of the same tool tooth, and its instantaneous cutting thickness model is shown in Figure 6b, and the equation of P_s_ point trajectory can be expressed Equation (17).
(17)$$\left\{\begin{array}{l} x_{s}=vt_{s}+r\sin\left(\omega t_{s}-\frac{2\pi j}{N}\right)+r_{0}\sin\left(\omega t_{s}+\psi \right)+A_{x}\sin\left(2\pi f_{x}t_{s}+\varphi_{x}\right)\\ y_{s}=r\cos\left(\omega t_{s}-\frac{2\pi j}{N}\right)+r_{0}\cos\left(\omega t_{s}+\psi \right)+A_{y}\sin\left(2\pi f_{y}t_{s}+\varphi_{y}\right) \end{array}\right.$$

The relationship between *t*s and *t* can be obtained by associating (13)–(17). The unique variable *t*s can be solved by MATLAB’s fsolve function. From this, it can be deduced that the instantaneous cutting thickness can be expressed Equation (18).
(18)$$h\left(\theta ,j\right)=L_{\mathrm{OB}}-L_{{\mathrm{OP}}_{s}}=r-L_{{\mathrm{OP}}_{s}}$$
where $L_{\mathrm{OB}}$ is the distance between the center point O of the milling cutter and the point P_s_; $L_{OP_{s}}\;$is the distance between the center point O of the milling cutter and the point P_s_, which can be expressed by Equation (19).
(19)$$L_{{\mathrm{OP}}_{s}}={\left[{\left(x_{s}-x_{o}\right)}^{2}+{\left(y_{s}-y_{o}\right)}^{2}\right]}^{\frac{1}{2}}$$

According to the above mechanism of forming the instantaneous cutting thickness in the 2D vibration-assisted micro-milling process, the calculation of the instantaneous cutting thickness is realized by numerical simulation using MATLAB computing platform. Since the eccentricity *r*_0_ of the milling cutter is 2 μm, Figure 7 shows the variation curves of the instantaneous cutting thickness with tooth position angle for vibration-assisted milling and conventional milling when the feed per tooth *f*_z_ = 10 μm/tooth, the amplitude *A_x_* = *A_y_* = 2 μm, and the frequency *f_x_ = f_y_* = 3000 Hz, as calculated by the above algorithm procedure.

By comparing the instantaneous cutting thickness of the milling cutter in conventional milling (CM) and vibration-assisted milling (VM) processes, it can be observed that the cutting thickness is no longer a smooth curve. This is due to the periodic contact or separation between the tool and the workpiece material, resulting in intermittent cutting of the tool. The actual cutting thickness oscillates with the change of tooth angle, and, in addition, due to the radial runout of the milling cutter, the cutting thickness generated by the two cutting edges of the milling cutter differs within a single revolution cycle, leading to long- and short-tooth cutting phenomena. Therefore, a cutting thickness model can more accurately reflect the real situation of cutting thickness in actual cutting processes.

## 4. Experimental Verification

### 4.1. Vibration-Assisted Micro-Milling System

The vibration-assisted micro-milling system used in this experiment mainly consists of a micro-milling system and a vibration module. The micro-milling system mainly consists of a three-axis horizontal micro-milling machine, spindle controller, built-in drive controller, micro-milling cutter, 3D force measuring instrument, data-acquisition card, and air compressor. The vibration module mainly consists of the signal generator, voltage amplifier, piezoelectric actuator, and vibration stage. The vibration-assisted micro-milling system is shown in Figure 8. One of the high-speed spindles of the micro-milling machine is the NAKANISHI BM320 high-speed spindle with a radial runout within 1 μm. The multi-axis built-in drive controller of the motion system is SURUGA SEIKI DS102 and the milling cutter in the experiment is an SGO Taiwan superfine tungsten steel-coated milling cutter. The force-measuring instrument model is KISTLER 9317B, its X and Y bi-directional sensitivity is 26Pc/N, the force-measuring range is ±200 N, and the data-acquisition card is ECON MI-7008.

The vibration module experimental setup is described as follows: the signal generator is a function/arbitrary waveform generator of model Agilent 33500B, which can output five basic waveforms and arbitrary waveforms, and the frequency range of arbitrary waveforms is 1 μHz~5 MHz. The piezoelectric driver is a Core Tomorrow PSt150/5 × 5/20 L, whose stiffness is 60 N/μm, applied voltage is 0~150 V, and capacitance is 1.6 μF.

In order to study the effect of different cutting directions and vibration parameters on the cutting force of bone material, vibration milling experiments of bone material were conducted with a fixed milling tool speed of 3000 r/min, a feed of 10 μm per tooth, and an axial cutting depth of 100 μm, and the initial phase of the signal source in X and Y directions were set to sin–cos, and the frequency and amplitude in X and Y directions were equal. In order to analyze the effect of cutting direction, frequency, and amplitude of 2D vibration on the milling force of bone material, using the single-factor experimental method, the rest of the cutting parameters remain unchanged, and each experiment only changes the cutting direction, frequency, or amplitude of a parameter so as to study the role of the variable. The specific cutting parameter settings are shown in Table 5. The amplitude and frequency of the experiments in groups 1, 2, and 3 are 0 Hz, indicating that the group is a conventional milling method without vibration.

### 4.2. Analysis of Experimental Results

The vibration-assisted micro-milling system used in this experiment mainly consists of a micro-milling system and a vibration module. The micro-milling system mainly consists of a three-axis horizontal micro-milling machine, spindle controller, built-in drive controller, micro-milling cutter, 3D force measuring instrument, and data-acquisition card.

#### 4.2.1. Cutting-Force Model Validation

The simulation study of the bone material milling force model was carried out using MATLAB software. The predicted curves of milling forces in X-direction and Y-direction for cutting along three directions, parallel, cross, and vertical, at amplitude *A* = 1 μm and frequency *f* = 3000 Hz, are shown in Figure 9, Figure 10 and Figure 11.

Considering that the process of collecting cutting force will be disturbed by noise signals during the experiment, a low-pass filter of 120 Hz is used to filter the original cutting-force signal, and the established cutting-force model is verified by the single-factor experimental method of cutting direction shown in groups 4, 5, and 6 in Table 5. The comparison between the model predicted cutting force and experimental cutting force for bone materials under different cutting directions is shown in Figure 9, Figure 10 and Figure 11. Comparing the predicted cutting-force curves of the model with the experimental cutting-force curves in different cutting directions, it can be seen that the predicted milling force model can better reflect the trend of the experimental cutting force and the alternating cutting phenomenon of long and short teeth due to the radial runout of the tool, but there is still a specific error in the magnitude of the cutting force, which is caused by two reasons: one, is that bone material as a biological material is non-homogeneous, and the strength of the bone unit is higher than that of the osteon. When the cutting edge of the tool is located at different cutting positions on the bone material, the cutting force will fluctuate within a certain range; secondly, as a hard and brittle material, bone material is prone to cracking or chipping during cutting, and the cutting uniformity is poor, which also makes the cutting force easily fluctuate within a certain range. In Figure 9, Figure 10 and Figure 11, when the tooth angle is 90° and 450°, the experimental data of cutting force is smaller than the simulation data, which may be due to the single-tooth cutting phenomenon caused by the large radial runout of the tool during the experiment.

The X-directional and Y-directional cutting forces *F_x_* and *F_y_* collected from each group of experiments were filtered to remove the interference signals. The RMS values of cutting forces in the three cutting directions were calculated. Their comparison with the RMS values of cutting forces predicted by the model is shown in Figure 12.

From Figure 12, it can be seen that the errors of RMS values of cutting-force prediction model in X and Y directions compared to the experiments are 10.56% and 6.57% for parallel cutting, 7.23% and 7.62% for cross cutting, and 6.48% and 11.59% for vertical cutting, respectively. Therefore, according to the analysis, the milling-force model considering the anisotropy of bone material can better predict the trend of dynamic cutting force of vibration-assisted micro-milling of bone material under different cutting directions. Figure 12 visualizes the comparison of the experimental cutting force RMS under the three cutting directions.

It can be seen from the figure that there are differences in the cutting force of bone material in three cutting directions, and the cutting force in vertical cutting is the largest among the three cutting directions, which is due to the vertical relationship between the direction of cutting speed and the orientation of the bone unit at the maximum cutting thickness, and the greater strength and fracture toughness of the bone material due to the fiber reinforcement of the bone unit, and the greater resistance to crack expansion and tool resistance, so that the cutting force generated when cutting bone material vertically is the largest. In the case of parallel cutting and cross cutting, the cracks in the cutting process expand along the boundary of the bone unit, and the resistance to crack expansion and tool resistance is smaller, and the cutting force is also smaller.

#### 4.2.2. Effect of Amplitude on Cutting Force

Based on the single-factor experimental method of amplitude shown in Table 5 for groups 1, 4, 7, 8, 9, 10, and 11, the milling experiments were performed on the bone material by fixing other parameters constantly and changing only the amplitude. The RMS values of *F_x_* and *F_y_* were calculated by filtering the X and Y cutting-force data collected from each group of experiments to remove the interference signals, as shown in Figure 13.

By analyzing the relationship between the cutting force and amplitude of the bone material, it is evident that vibration application leads to a decreasing-then-increasing trend in the cutting force with increasing amplitude. This can be attributed to various factors. As the amplitude increases, the net cutting time in each cutting cycle decreases, resulting in increased empty cutting time, enhanced tool–workpiece separation effect, and reduced effective frictional contact area between the tool and bone material. These factors collectively contribute to the reduction in cutting force. Furthermore, increased amplitude facilitates chip breaking and reduces chip friction, thereby reducing the milling force to a certain extent. However, continued increase in amplitude leads to an increase in the thickness of the cutting layer caused by the impact, resulting in an upward trend in the milling force. At a fixed frequency of 3000 Hz, the cutting force reaches its minimum value at an amplitude of 2 μm. Moreover, compared to conventional milling, the RMS values of the cutting force in the X and Y directions are reduced by 9.44% and 12.81%, respectively.

#### 4.2.3. Effect of Frequency on Cutting Force

Milling experimental studies on bone materials were performed based on the frequency single-factor experimental method shown in Table 5 for groups 1, 8, 12, 13, and 14. The RMS values of *F_x_* and *F_y_* were calculated by filtering the X and Y cutting-force data collected from each group of experiments to remove the interference signals, as shown in Figure 14.

By analyzing the relationship curve between cutting force and frequency of bone material, it can be seen that after the application of vibration, the cutting force shows a trend of gradual decrease with the increase in frequency, which is mainly because with the increase in frequency, the frequency of contact and separation between the tool and the workpiece increases, and the frequency of decrease in milling force per unit time also increases, which leads to the reduction of milling force. In addition, in the process of the cutting force changing with amplitude and frequency, as seen in Figure 13 and Figure 14, the experimental value is slightly lower than the predicted value. This may be because the 2D vibration-assisted milling method changes the chip-formation process of the bone material, and the higher-frequency impact vibration increases the strain rate of the bone material during the cutting process. This effect allows the crack to penetrate the bone with minimal cutting energy and expand in the main shear direction. Additionally, the chip is more prone to fracture under the action of high-frequency impact, resulting in a smaller chip curvature. Moreover, the average friction between the tool and the material decreases, leading to a reduction in cutting force. At a fixed amplitude of 1.5 μm and a frequency of 4000 Hz, the RMS values of cutting forces in the X and Y directions decreased by 11.60% and 14.08%, respectively, compared to the conventional milling method without vibration.

## 5. Conclusions

In this paper, the vibration-assisted micro-milling technique is applied to the cutting of bone materials, and the mathematical modeling and experimental study of the cutting forces of bone materials in the vibration-assisted micro-milling process are completed, and the main findings are as follows:(1)The mechanical properties of bone materials were analyzed. The stress–strain curves of bovine cortical bone samples were obtained by shear tests in vertical, parallel, and cross directions, and the shear strength of bone materials in different directions was calculated according to the test results. The analysis results show that the mechanical properties of bone materials in the three directions are different, and the shear strength in the parallel and cross directions is small. In the vertical direction, the bone material exhibits linear elasticity and elastoplasticity, and the shear strength is large. Therefore, bone materials have obvious anisotropy;(2)Based on the mechanical analysis method and the oblique cutting micro-element model, the cutting-force model of vibration-assisted micro-milling bone material with multiple processing parameters is established by considering the mechanical properties of bone material anisotropy, tool geometric parameters, ploughing effect, and the kinematics characteristics of the tool in the two-dimensional vibration-assisted micro-milling process. The variation law of cutting force is simulated by MATLAB. It can be seen that after vibration is applied, the change of cutting force with tooth-position angle is a constantly oscillating curve. Because the shear strength is the largest when cutting along the vertical direction, the cutting force in this direction is also the largest; due to the radial runout of the tool, the peak cutting force generated by the two cutting edges is different;(3)The vibration-assisted micro-milling experiment of bone materials was completed, and the correctness of the cutting-force prediction model was verified. The effects of cutting direction and vibration parameters on cutting force were studied. The results show that among the three cutting directions, the cutting force generated by bone materials in the vertical cutting direction is the largest, and the cutting force of bone materials in the parallel cutting direction and the cross-cutting direction is close. After the vibration effect is applied, the cutting force decreases first and then increases with the increase in the amplitude. With the increase in frequency, the cutting force decreases gradually. In the process of vibration-assisted micro-milling of bone materials, selecting the appropriate cutting direction and vibration parameters can achieve the effect of reducing the cutting force.

## Figures and Tables

**Figure 1 micromachines-14-01422-f001:**
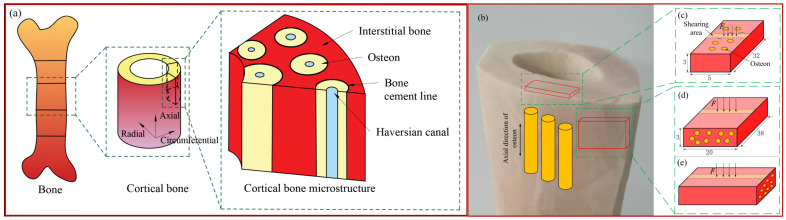
Structure, size, and loading direction of bone material samples: (**a**) structure diagram of bone material; (**b**) bovine femur diaphysis; (**c**) parallel samples; (**d**) vertical samples; and (**e**) across samples.

**Figure 2 micromachines-14-01422-f002:**
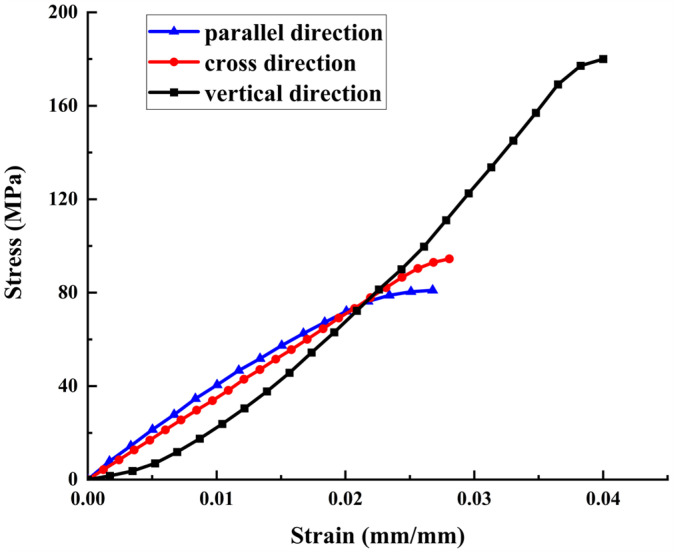
Shear stress–strain curves of bone material along different directions.

**Figure 3 micromachines-14-01422-f003:**
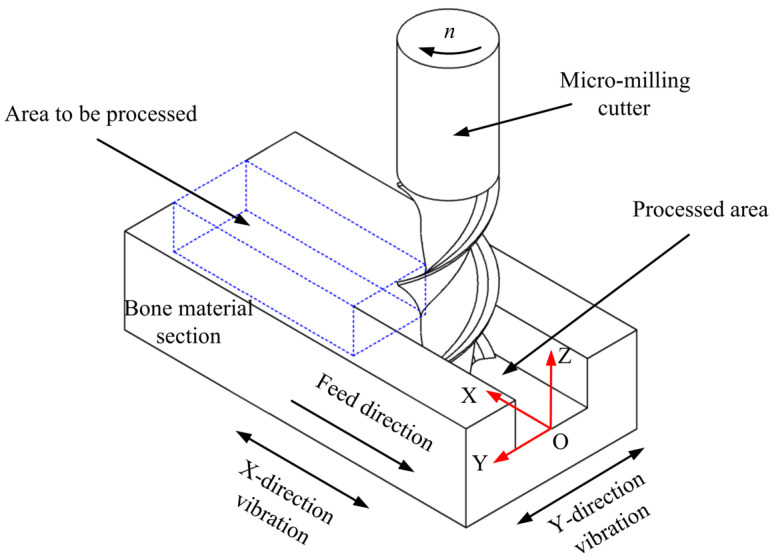
Schematic diagram of vibration-assisted milling of bone materials.

**Figure 4 micromachines-14-01422-f004:**
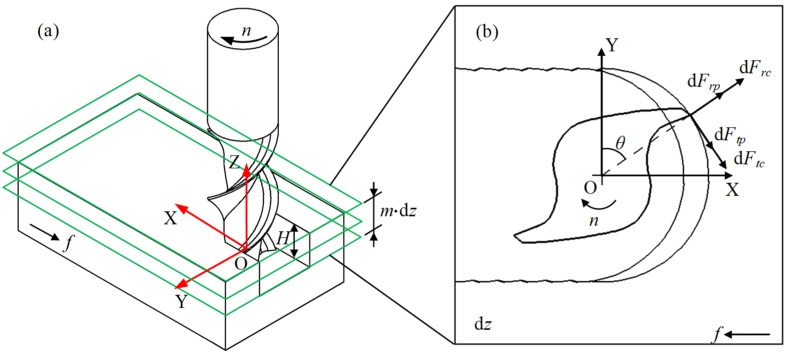
Schematic diagram of the cutting-force model of milling bone material (**a**) discrete method of cutting sheet layer; and (**b**) force situation of bone material.

**Figure 5 micromachines-14-01422-f005:**
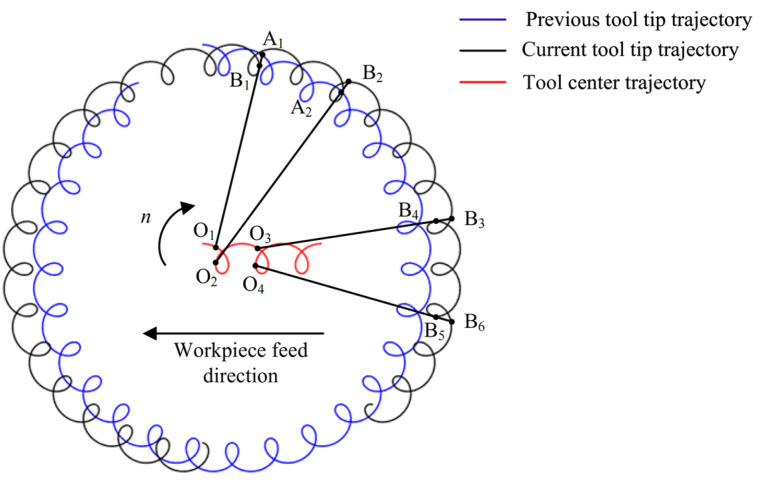
Schematic diagram of instantaneous cutting thickness formation of bone material.

**Figure 6 micromachines-14-01422-f006:**
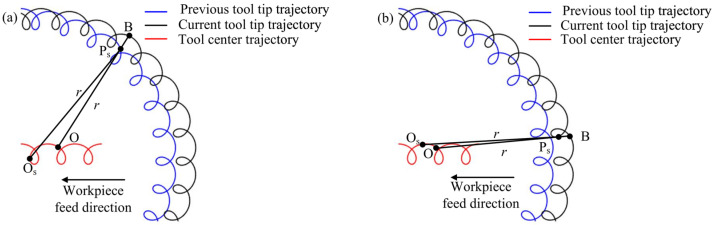
The instantaneous cutting thickness calculation method (**a**) caused by different tool tip and (**b**) caused by the same tool tip.

**Figure 7 micromachines-14-01422-f007:**
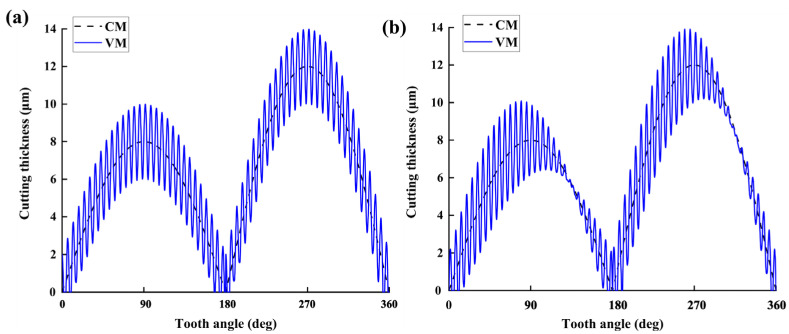
Cutting thickness curve of vibration-assisted milling of bone material when (**a**) signal source is sin–cos and (**b**) signal source is sin–sin.

**Figure 8 micromachines-14-01422-f008:**
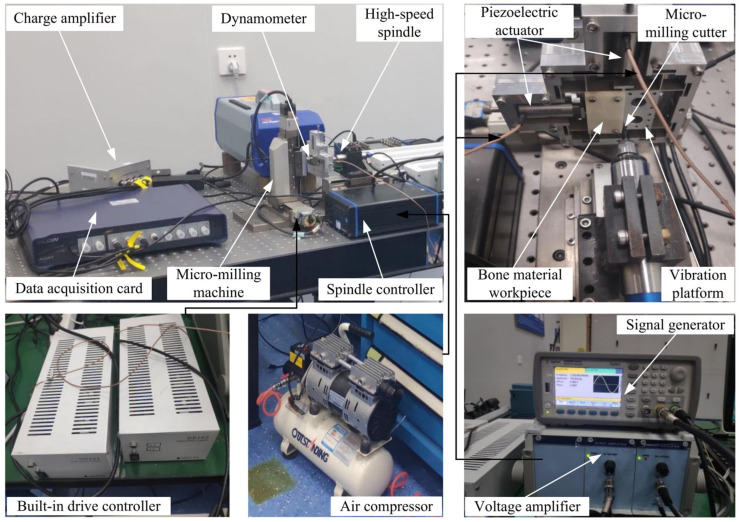
Vibration-assisted micro-milling system.

**Figure 9 micromachines-14-01422-f009:**
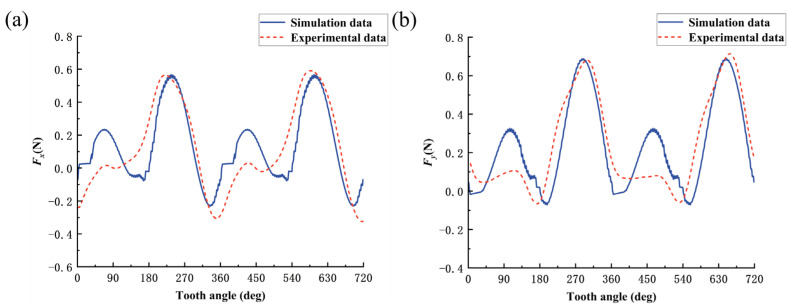
Comparison between predicted and experimental values of milling force in parallel cutting mode: (**a**) cutting force in X direction; and (**b**) cutting force in Y direction.

**Figure 10 micromachines-14-01422-f010:**
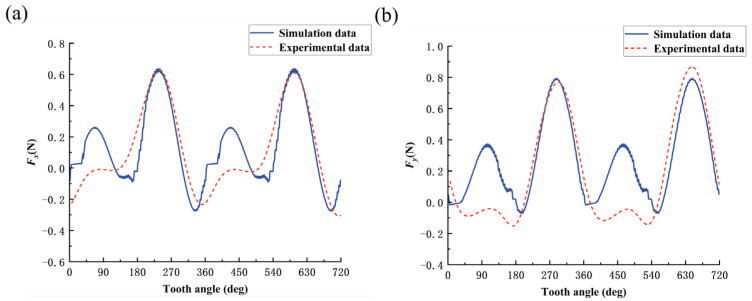
Comparison between predicted and experimental values of milling force in across cutting mode: (**a**) cutting force in X direction; and (**b**) cutting force in Y direction.

**Figure 11 micromachines-14-01422-f011:**
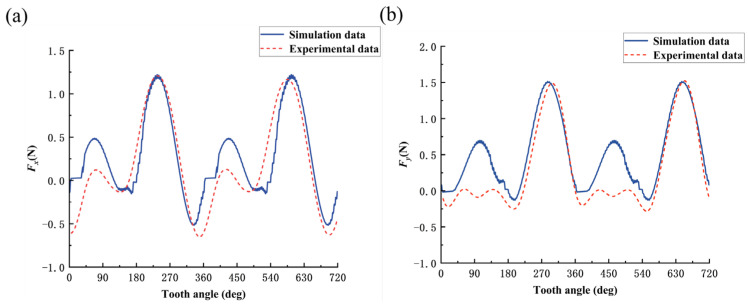
Comparison between predicted and experimental values of milling force in vertical cutting mode: (**a**) cutting force in X direction; and (**b**) cutting force in Y direction.

**Figure 12 micromachines-14-01422-f012:**
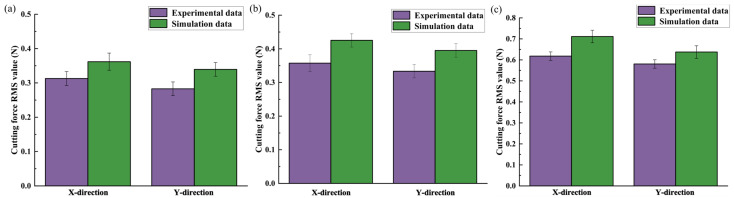
Comparison of RMS value of cutting force in three cutting methods: (**a**) parallel cutting direction; (**b**) cross cutting direction; and (**c**) vertical cutting direction.

**Figure 13 micromachines-14-01422-f013:**
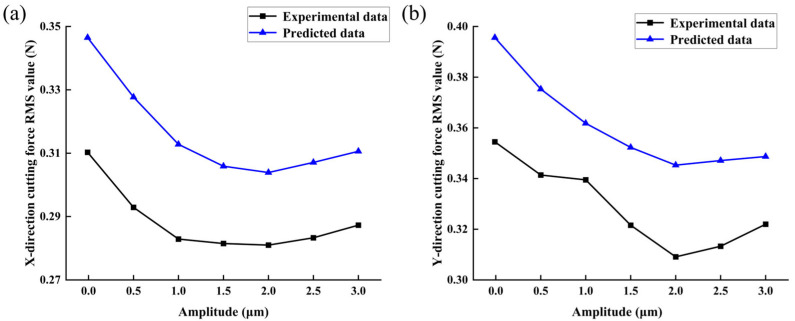
Effect of vibration amplitude on milling force: (**a**) in X direction; and (**b**) in Y direction.

**Figure 14 micromachines-14-01422-f014:**
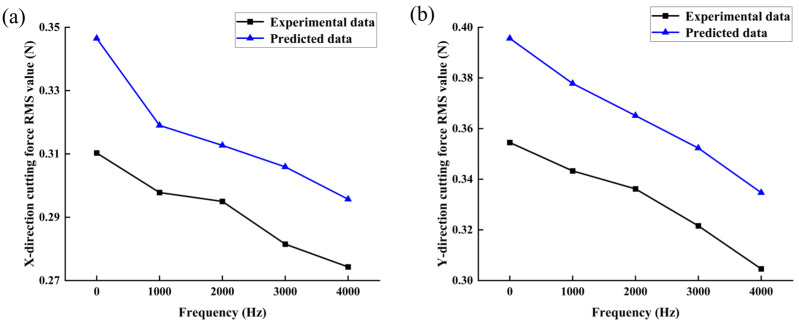
Effect of vibration frequency on milling force: (**a**) in X direction; and (**b**) in Y direction.

**Table 1 micromachines-14-01422-t001:** Anisotropic shear strength of bone material.

Load Direction	Shearing Strength (MPa)
parallel direction	81.03
cross direction	94.48
vertical direction	180.12

**Table 2 micromachines-14-01422-t002:** Integral limit of spiral milling tool.

Cutting Area	Rotation Angle	Point Range	
		Lower Limit Angle *θ_st_*	Upper Limit Angle *θ_ex_*
access to cutting	0 ≤ *θ* ≤ *θ_z_*	*0*	*θ*
continuous cutting	*θ_z_* < *θ* ≤ *θ_d_*	*θ–θ_z_*	*θ*
exit cutting	*θ_d_* < *θ* ≤ *θ_d_ + θ_z_*	*θ–θ_z_*	*θ_d_*

In the table: $\theta_{d\;}$ is the maximum milling angle, for two-blade flat-end milling cutter, $\theta_{d}=\pi$.

**Table 3 micromachines-14-01422-t003:** Cutting performance parameters of bone materials in various cutting modes [25].

Cutting Direction *i*	Friction Coefficient *μ_i_*
1	0.82
2	0.98
3	0.85

**Table 4 micromachines-14-01422-t004:** Geometric parameters of the spiral milling machines.

Micro- Milling Cutter Type	Spiral Milling Cutter Geometry Parameters					
	Diameter d (μm)	Rake Angle $\gamma_{n}$ (deg)	Back Angle $\alpha_{n}$ (deg)	Spiral Angle λ (deg)	Radius of Obtuse Circle $r_{e}$(μm)	Tool Eccentricity $r_{0}$(μm)
SGOS550 0.8 × 2.4 C × 4 D × 50 L	800	5	10	35	5	2

**Table 5 micromachines-14-01422-t005:** Single-factor experimental processing parameters for bone milling.

Experiment Number	Cutting Direction *i*	Amplitude *A* (μm)	Frequency *f* (Hz)
1	1	0	0
2	2	0	0
3	3	0	0
4	1	1	3000
5	2	1	3000
6	3	1	3000
7	1	0.5	3000
8	1	1.5	3000
9	1	2	3000
10	1	2.5	3000
11	1	3	3000
12	1	1.5	1000
13	1	1.5	2000
14	1	1.5	4000

## Data Availability

Not applicable.

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
