# Peer review of "Cutting-Force Modeling Study on Vibration-Assisted Micro-Milling of Bone Materials"

_micromachines, 2023, doi:10.3390/mi14071422_

Round 1
Reviewer 1 Report
Overall, the paper is well-structured, and the results are convincing. The quality of this paper, in my opinion, is worthy of publication. The following comments are mostly for clarification.
1. There is a problem with the expression of the sentence in line 79.
2. Is there any clinical reference for the loading speed of 0.5 mm/min in line 126? Please clarify in the text.
3. Please describe the structure of the bone sample in detail in Figure 1 to facilitate understanding.
4. It is suggested to include error bars in Figs. 12 and 13 to clarify the reproducibility of the experimental results.
5."Experimental investigation of material removal in elliptical vibration cutting of cortical bone" can provide a reference for the section of introduction about elliptical vibration cortical bone cutting.
The the Quality of English Language of paper is good and easy to be understanded.
Reviewer 2 Report
The comments of the reviewer on micromachines-2494012:
The aim of this study is to increase surgical safety and facilitate patient recovery by examining vibration-assisted micro-milling technology to remove bone material. The main goal is to reduce the cutting force and improve the surface quality. First, a predictive model was developed to estimate the cutting force during two-dimensional milling using the vibration of bone materials. This model takes into account the anisotropic structural characteristics of bone material and the kinematics of the grinding tool. Subsequently, an experimental platform is established to validate the shear force model for bone materials. Micro-grinding experiments are performed on bone materials by changing the cutting direction, amplitude, and frequency to evaluate their effect on the cutting force. The experimentally observed data reveals that the selection of appropriate machining parameters can effectively minimize the cutting force in fine grinding of bone materials with the help of two-dimensional vibration.
This is a well-written paper and can be recommended for publication in “Micromachines-MDPI” after the following issues could be rationally responded to by the authors:
- What are the main novel aspects/originalities of the present works that discriminate that from similar studies?
2. No attempts have been made for mathematical modeling of bone structure (i.e., on the basis of the poroelastic deformation and poroelasticity). How the porosity and density of the solid skeleton could influence the shear and Young’s modulus as well as the whole mechanical response of the bone material subjected to micro-milling? Please clarify within your manuscript as well.
3. In general, the mechanical behavior of the bone material is said to be “anisotropic”; however, the more specific type of anisotropy (i.e., orthotropic, transversely anisotropic, and so on) has been not displayed. Please explain that in more detail in your paper.
4. The role of the porosity in the introduced shear force coefficients in Eq. (6) has been explained and discussed. In the light of comment#2, please respond to this query.
5. In all directional cutting modes (parallel, across, and vertical), some noticeable discrepancies between the maximum values of the first and third peaks based on the simulation data and those of the experimental study are detectable. How the differences between these two can be interpreted (i.e., what are the main reasons for the occurrence of these major discrepancies at \theta=90 and 450o)?
6. The plots of X- and Y-directional cutting force as a function of amplitudes reveal some meaningful differences between the predicted data by numerical simulations and those of experimentally observed data. How these discrepancies can be explained? Could the stiffness/rigidity of the cutter be a reason? Please clarify that with more detail in your manuscript.
7. Since the medium of bone is poroelastic and it is vibrated by the micro-milling force, the following theoretical works on the mechanical responses of poroelastic structures due to a moving load can be of interest to the readers:
https://www.sciencedirect.com/science/article/abs/pii/S002074031500065X
https://link.springer.com/article/10.1007/s00419-004-0349-2
8. The dynamic interactions between the cutter and the bone material during micro-milling have been not taken into account. How consideration of these interactions could influence the plots of the cutting force-amplitude in various cutting directions as well as other obtained results, particularly those based on the vibration-assisted milling? Please clarify.
9. In Fig. 13, the y-label of Fig. 13(b) should be revised to “Y-direction cutting force …” since the corresponding plotted results are related to the y-directional cutting force of the micro-milling process. Please recheck and make the required modifications.
10. In the presented graphs in Figures 2, 7, and 9-11, please use dashed, dotted-dashed, and solid lines to discriminate the results instead of using different colors since in the printed version, which is commonly based on the black-white format, these graphs cannot be properly distinguished.
11. The English of the paper should be improved.
After the above-raised issues are rationally replied to by the authors, I can recommend the paper for publication in the esteemed journal of “Micromachines”.
Some English revisions are required. Please also see the above-provided comments.
